# Association between Motivation in Physical Education and Positive Body Image: Mediating and Moderating Effects of Physical Activity Habits

**DOI:** 10.3390/ijerph20010464

**Published:** 2022-12-27

**Authors:** Rasa Jankauskiene, Danielius Urmanavicius, Migle Baceviciene

**Affiliations:** 1Institute of Sport Science and Innovations, Lithuanian Sports University, 44221 Kaunas, Lithuania; 2Department of Physical and Social Education, Lithuanian Sports University, 44221 Kaunas, Lithuania

**Keywords:** body appreciation, self-determination, physical education classes, support of autonomy, physical activity, habits, adolescents

## Abstract

Concerns about body image might prevent adolescents from participating in physical education (PE) classes and physical activities during leisure-time. In this cross-sectional study, we assessed the relationships between teacher support of autonomy, student motivations for PE, and positive body image, in a sample of Lithuanian adolescents. A total of 715 adolescents (51.89% girls) participated in the study. Ages ranged from 14 to 18 years, with a mean age of 16.00 (SD = 0.79) for girls and 15.99 (SD = 0.75) for boys. The questionnaire consisted of demographic questions, the Learning Climate Questionnaire, the Revised Perceived Locus of Causality in Physical Education Questionnaire, the Body Appreciation Scale-2, the Self-Report Habit Index for Physical Activity (PA), perceived physical fitness (PPF), and Godin Leisure-Time Exercise Questionnaire. The results showed that teacher support for students’ autonomy was associated with higher positive body image. In adolescent girls, autonomous motivation for PE was the mediator between teacher support of autonomy and positive body image. The associations between self-determined motivation in PE and positive body image were mediated by PPF on one hand, and through PA habits and PPF on the other hand of the structural equation model (in girls). PA habits moderated associations between PE motivation and PPF. Associations between PE motivation and PPF were stronger in girls with the lowest PA habits compared to girls with mean PA habits. These results suggest that PE classes are important for promoting students’ positive body images. PE teachers are advised to use pedagogical strategies that promote the self-determined motivation of students for PE. Increased self-determined motivation for physical education might be an effective strategy for physical self-perception and positive body image promotion in adolescent girls, especially those with low PA habits.

## 1. Introduction

Participation in sports and moderate to vigorous physical activities is related to a number of physiological and psychological benefits [1,2]. School-age children and adolescents should accumulate an average of one hour of moderate-to-vigorous aerobic physical activity (PA) per day, and regularly do strength-training exercises, while avoiding activities related to a sedentary lifestyle [3]. However, a large number of studies report that 80% of adolescents do not meet these recommendations [4,5,6]. Studies show a strong decline in PA during the transition from childhood to adolescence [5]. Physical education (PE) is an important school subject that leads to the growth of further experience, and, where educative elements are ongoing, in relation to the future [7]. Educative PE is not about doing sports, fitness instruction, PA promotion or obesity prevention, it is about the joint personal and physical growth of teachers and students through sports and PA [7]. Physical competence and confidence in the physical-self are important aspects of human growth and self-worth [8], and therefore, gaining more knowledge about the PE–related factors that promote these outcomes is important for science and practice.

PE is a school subject in which the body is the focus of curricular outcomes. In PE, the body is displayed for social comparison, and is judged for its performance [9]. Public attention to students‘ bodies might be a source of stress for some students, especially those with lower physical competencies [10]. There is some evidence that a student‘s satisfaction with the body is related to the greater PA and perceived sport-promoting effect of PE [11,12]. Conversely, body dissatisfaction in PE decreases an adolescent‘s satisfaction and enjoyment of PE, and lowers achievements in it [9,13,14]. Based on self-determination theory (SDT) [15], previous studies showed that satisfaction with body image is associated with more intrinsic exercise motivation [16,17], but few empirical studies have investigated whether and how motivational processes in PE are associated with the perception of a student‘s physical self, and specifically positive body image. The possible motivational mechanisms underlying these associations are not well understood. Analysing these mechanisms might lead to the development of evidence-based pedagogical strategies that support the development of positive body image through PE.

### 1.1. Body Image, Physical Activity, and Physical Education

Body image is an individual’s subjective view of their body, including perceptual and attitudinal components [18]. The attitudinal component includes global satisfaction or dissatisfaction with appearance, and cognitive, affective, and behavioural elements [19]. Body weight or appearance dissatisfaction is a central aspect of negative body image, and it is associated with negative psychological and physical outcomes, including lower PA [20,21] and dysfunctional exercise [22]. Scholars recently conceptualised the embodiment as a complex, moving beyond the study of appearance to body functionality, and started to explore factors that protect body image from sociocultural pressures [23,24,25]. The concept of positive body image was developed on the basis of positive psychology. Positive body image is described as a multifaceted construct that includes an appreciation of the body and its functional abilities, an awareness of the body‘s needs, and the ability to resist and protect oneself from sociocultural pressures regarding appearance [23,26,27]. There is ongoing discussion about whether positive body image is an independent construct or one extreme of the continuum of negative body image [28]. A recent meta-analysis showed that positive body image (operating as body appreciation) was inversely associated with numerous indices of eating and body image disturbances, and general psychopathology (depression, anxiety), and positively with adaptive well-being constructs such as self-esteem and self-compassion. These associations were still evident after controlling for the influence of negative body image, suggesting that positive body image is an independent construct but not the opposite of negative body image [29].

#### 1.1.1. Body Image and Physical Activity

Research shows that physically active adolescent boys and girls demonstrate fewer body image concerns and greater positive body image [11,30]. An analysis of PA-based interventions concluded that these interventions, especially when implemented in school settings, are beneficial for developing positive physical self-perception in adolescents, and especially for the perceptions of sport competence and physical fitness [31]. According to the hierarchical exercise and self-esteem model (EXSEM) [8], the possible impact of PA on positive physical self-perception might be explained by improved self-efficacy (or confidence in one’s abilities) and improved perceptions of physical competence, physical fitness, and physical attractiveness, which further increase global self-perception. The results of previous studies suggest that positive body image (body appreciation) is an important mediator between leisure-time PA, perceived physical fitness (PPF) and global self-esteem in adolescent girls and boys [32]. Increased body functionality, or valuing functional body features over appearance, might also explain how PA increases positive body image [25]. A previous study showed that adolescent girls participating in sports reported higher body functionality-related attitudes compared to sedentary adolescent girls [33]. Valuing body function over appearance is an important variable through which positive body image might be increased in interventions for young adult women [25].

There is evidence that an affective dimension of body image, which includes a range of self-conscious emotions such as body-related guilt, shame, embarrassment, and envy, is associated with PA. The results of a recent longitudinal study suggested that negative body-related emotions such as guilt, shame, and embarrassment increased with age in adolescent girls and boys, and were associated with lower PA [34]. Conversely, body-related pride (a positive self-conscious emotion that presents individual behaviours or characteristics that are congruent with the ideal self or socially desirable) was positively associated with PA in boys and girls [34]. Another longitudinal study showed that negative changes in body-related emotions predicted lower sports experience and dropout from sports in adolescent girls [35]. Overall, these findings suggest that positive body image-related emotions might help to keep adolescents in PA, and to protect them from dropout.

#### 1.1.2. Body Image and Physical Education

A significant proportion of adolescents experience body image concerns in PE [36,37,38]. Body satisfaction is one of the most important components affecting how students perceive PE classes and whether they experience them as stimulating and encouraging leisure-time PA [12]. There is evidence that body dissatisfaction is associated with a more negative perception of physical competence in PE [38]. Predominantly qualitative literature suggests that PE might activate events that are likely to heighten body image concerns in adolescent boys and girls [9,39,40,41]. Some students equate PE with team sports, and adolescent girls with lower physical competence see this as a barrier to participation in PE [42]. The qualitative study showed that body visibility in PE might be stressful for some students [37]. A qualitative study with a Norwegian sample of adolescents showed that PE is more stressful for students with lower competence in sports, and a more negative body image [10]. There is, however, a lack of empirical studies exploring body image in PE. In their recent review, Kerner et al. [9] argue that body image problems in PE are so greatly overlooked and ignored in research and practice that “body image in PE might appear to be like the ’elephant in the room’”. We thus aimed in the present study to provide more empirical knowledge on this issue.

### 1.2. Self-Determination Theory in Physical Education and Body Image

Previous studies have demonstrated that exercise only has a positive effect on body image when there is autonomous motivation for exercise [17]. The self-determined theory (SDT) of human motivation distinguishes between different forms of motivation, such as controlled and autonomous regulations [15,43]. The internalisation of self-determination, or autonomous motivation, fulfils three basic psychological needs (BPN): autonomy (the need to feel that one‘s behaviour is self-determined and the reasons for action are self-endorsed), competence (the need to feel effective and capable of performing various tasks) and relatedness (the feeling of connectedness with others) [15]. Research suggests that support for BPN is associated with greater autonomous motivation for PA or PE [44,45]. SDT describes types of motivation in a continuum of behavioural regulations, from amotivation, through external motivation, to self-determined motivation. Amotivation means that a person lacks intentionality for the behaviour. Controlled forms of external motivation are external regulation (a person performs activities under pressure from external agents) and introjected regulation (activities are performed to avoid internal pressures such as guilt and anxiety). More autonomous types of external motivation include identified regulation (behaviour is personally important, a person experiences high volition to act) and integrated regulation (activities assimilate with personal values and interests). Finally, autonomous or self-determined regulation means that a person acts because of personal interests, pleasure, and other self-endorsed reasons [15].

In their meta-analysis, Vasconcellos et al. [44] showed that more autonomous forms of regulation (intrinsic motivation, identified regulation, and introjected regulation) are positively associated with the affective, behavioural, and cognitive outcomes of students in PE. Other studies in the exercise-related domain suggest that self-determined motivational regulations predict more adaptive behavioural, cognitive, and physical self-evaluation patterns than external regulation [46,47,48]. A recent longitudinal study observed less decline in the PA of adolescents who maintained high autonomous motivation together with high enjoyment goals compared to other adolescents [49]. Self-determined motivation is thus an important agent of students‘ PA in PE, and in the maintenance of PA outside school [50].

Teaching might be supportive of autonomy, or controlling. Autonomy-supportive teaching means that teachers use non-controlling language, try to understand student perspectives, support students in their choices, provide meaningful tasks, share responsibilities, and so on. These pedagogical strategies satisfy the BPNs of students [44]. Fulfilling a student‘s BPNs further leads to more autonomous PE motivation, which is further associated with positive outcomes such as greater concentration, greater enjoyment in PE and higher self-esteem [44,51,52,53,54].

Previous findings suggest that an adolescent‘s negative body image (operating as social physique anxiety) negatively predicts autonomous motivation, and positively predicts low-quality motivation for PE and greater avoidance of PE classes [36]. Body dissatisfaction in PE decreased adolescent satisfaction and enjoyment of PE, which are the main features of intrinsic motivation [9,13,14]. A recent study by Kerner, et al. [38] showed that body satisfaction during PE was associated with the perceived sport-promoting role of PE in adolescent girls, and associations between body satisfaction during PE and the perceived sport-promoting role of PE (being physically active outside school) were mediated by autonomous motivation in both genders. While studies have shown that satisfaction with body image is associated with more intrinsic exercise motivation [16,17], only a few studies have empirically investigated whether and how teacher behaviours and motivational processes in PE are associated with positive body image in students as an outcome of autonomous PE motivation. Motivation has been considered an antecedent rather than a consequence of body image outcomes, and exercisers who are intrinsically motivated are less focused on how their bodies appear to others [46].

### 1.3. Physical Activity Habits as Mediators and Moderators between Autonomous Motivation for Physical Education and Positive Body Image

#### 1.3.1. Physical Activity Habits and Autonomous Motivation

One of PE‘s long-time tasks is promoting PA as habitual behaviour. PA habits have an important place in the maintenance of PA behaviour [55]. Modern theory describes habits as a specific action or behavioural tendency that is enacted with little conscious awareness in response to a specific set of associated conditions or contextual cues [55,56]. Habits include multiple actions that are based on deliberate or automatic control [57]. As with other habits, PA habits originate in goal pursuit, because individuals tend to repeat behaviours that are rewarding [58]. It has been proposed, however, that individuals become less sensitive to previous motives when behaviour becomes habitual [57]. It has been suggested that habitual behaviours proceed without high cognitive efforts, and might be performed in conditions when self-control and motivation are low [59,60]. There is evidence that PA habits develop more quickly if PA is driven by self-determined motivation [58,61].

#### 1.3.2. Physical Activity Habits as a Mediator between Physical Education Motivation and Positive Body Image

The trans-contextual model of motivation (TCM) explains how motivation is transferred from one context to another [62]. TCM consists of three theories of motivation: SDT, the theory of planned behaviour [63], Vallerand‘s hierarchical model of intrinsic, and extrinsic motivation [64]. A growing body of research shows that PE environments that support the self-determined PE motivation of students also promote autonomous motivation and behaviour in other contexts, such as leisure time PA [50,52,65,66]. In the present study, we thus assumed that PA habits might be an outcome of autonomous motivation for PE. Since previous studies demonstrated that PA is associated with elevated levels of autonomous motivation and positive body image, we hypothesised that PA habits might be a mediator between these two things.

#### 1.3.3. Physical Activity Habits as Moderators between Physical Education Motivation, Perceived Physical Fitness, and Positive Body Image

Individuals differ in the extent to which they experience their PA as habitual [57]. PA habits might moderate associations between PE motivation and positive body image. According to the literature, strong PA habits might be associated with high autonomous motivation and positive body image in adolescents. Further, strong PA habits might protect adolescents from the negative effect of low autonomous motivation for PE on positive body image. Specifically, strong habits might persist in contexts where motivation might be low [60], and adolescents with strong PA habits, but low motivation for PE, might still have a positive body image as an outcome of higher PA. Conversely, adolescents with weak PA habits might benefit from increased teacher support of autonomy and increased motivation for PE more than adolescents with strong habits, since they have a wider margin for improvement. An analysis of PA interventions regarding the physical self-concept of adolescents showed that greater effects were achieved for those with lower initial levels of PA and physical self-perceptions [31]. We thus aimed to test these assumptions and expand our knowledge of this subject.

### 1.4. The Present Study

The aim of this study was to test the SDT-based hypothetical motivational model of teacher support of autonomy, student motivation for PE, and positive body image (operating as body appreciation). An additional objective was to assess the mediating effects of PA habits and PPF between motivation for PE and positive body image. Finally, we developed a second objective to test the moderating effects of PA habits between motivation for PE, PPF, and positive body image. According to SDT, we expected that teacher support of autonomy would be associated with greater adolescent autonomous motivation for PE, and that the latter would be associated with higher body appreciation (Figure 1). We hypothesised that PA habits and PPF would mediate associations between autonomous motivation for PE and body appreciation. Finally, we expected that the associations between

## 2. Materials and Methods

### 2.1. Study Participants and Procedure

The convenience sample of this cross-sectional study consisted of 715 adolescents (51.89% of whom were girls) from 13 schools representing five cities/towns, and their rural areas, in Lithuania. The study was conducted in 2022 June. The schools were contacted based on researchers’ personal contacts and previous collaborations, and participated in the study voluntarily. The only inclusion criterion applied—the study participants had to attend general profile public school, the 9th or the 10th grade. The average age of the sample was 16.00 (SD = 0.79) for girls and 15.99 (SD = 0.75) for boys, and the age range was 14 to 18 years.

The study was approved by the Institutional Social Research Ethics Committee (approval No. SMTEK-113). The research was organised in line with the Declaration of Helsinki’s ethical and legal principles. The Strobe Statement for cross-sectional studies was followed in the present study.

Permission to implement the study was obtained from the school’s directors. Information about the study was disseminated to parents via electronic sources, and their written consent was obtained. The online survey link was circulated with the help of the seven previously instructed school’s PE teachers, and questionnaires were filled in during theoretical PE classes. The survey was administered and implemented using the SurveyMonkey platform. Students had the option to agree or refuse to participate in the study through the question: “Do you agree to participate in this study?” If they declined to participate, the survey was terminated. There was also an option to stop participation in the study at any other point by closing the browser without recording the answers already provided. The survey took approximately 25–35 min to complete, with no time limit. The questionnaire consisted of several sections, including demographic information, questions about learning climate, motivation for PE, PA habits, PPF, and positive body image. All questions were mandatory; therefore, the final dataset contained no missing cases.

### 2.2. Study Measures

Perceived teacher support of autonomy was measured using the Lithuanian version of the Learning Climate Questionnaire (LCQ) [67]. This questionnaire is typically used in specific learning settings, such as a particular class, at the college or graduate school level. LCQ measures the degree to which students perceive their teachers as reinforcing their autonomy. A sample item is “I feel that my teacher provides me with choices and options”. Participants indicated the extent to which they agreed with the statements on a 7-point Likert scale (1 = Strongly Disagree, 4 = Neutral, 7 = Strongly Agree). The negatively formulated item No. 13 was reversely recorded, and all the response options were averaged. A higher score on the scale represents a higher perception of the autonomy-supported classroom environment. The adequate psychometric properties of the Lithuanian version of the questionnaire were previously confirmed [68]. There was high internal consistency in the current sample (α = 0.96).

Motivation for PE was measured by the Lithuanian version of the Revised Perceived Locus of Causality in PE scale (PLOC-R) [69]. PLOC-R is a 19-item, self-report instrument based on SDT that measures student motivation for PE. The instrument has five subscales: amotivation, external regulation, introjected regulation, identified regulation, and intrinsic motivation. Participants respond to each question using a 7-point Likert scale ranging from 1 (totally disagree) to 7 (totally agree). Participants were asked to respond to the items using the stem: “I take part in PE…”, followed by different reasons: “But I really don’t know why” (amotivation); “Because in this way I will not get a low grade” (external regulation); “Because I would feel bad if the teacher thought that I was not good at PE” (introjected regulation); “Because it is important to me to do well in PE” (identified regulation); “Because PE is enjoyable” (intrinsic motivation). For this statistical analysis, we only used the Relative Autonomy Index (RAI) calculated by the equation: (−3 × amotivation) + (−2 × external regulation) + (−1 × introjected regulation) + (1 × identified regulation) + (2 × intrinsic regulation). Subscale product scores are summed to indicate the level of self-determination. A higher score indicates a higher level of autonomous motivation for PE classes [70]. The Lithuanian version of the questionnaire demonstrated acceptable psychometric properties [68].

Positive body image was measured by the Lithuanian version of the Body Appreciation Scale-2 (BAS-2) [71]. A unidimensional 10-item scale was designed to measure body acceptance, body appreciation, and resistance to pressure from the media’s ideals of appearance. An example item of BAS-2 is as follows: “I feel like I am beautiful even if I am different from media images of attractive people (e.g., models, actresses/actors).” Response options are rated on a five-point Likert-type scale (1 = Never, 5 = Always). Higher scores indicate better body appreciation. The Lithuanian translation of this instrument demonstrated adequate psychometric properties and a unidimensional factor structure [72]. The internal consistency of the scale in the present study was α = 0.95.

The strength of PA habits was measured by the Lithuanian version of the Self-Report Habit Index (SRHI) [56]. The SRHI can be used to evaluate a wide range of behaviours, such as snacking, watching TV, and eating sweets. In this study, the SRHI was adapted for PA. The SRHI consists of a stem (“Behaviour x is something…”) that is adapted for different behaviours (e.g., “Physical activity is something…”), followed by 12 items with 7-point Likert response options. An example item is “Physical activity is what I do frequently”. The adequate psychometric properties and unidimensional factor structure were previously confirmed in the Lithuanian general population [73]. The Cronbach alpha of this instrument was good in the present study (α = 0.90).

Perceived physical fitness (PPF) was assessed by only one self-developed question “How do you evaluate your own fitness level compared with others?” Participants could choose an answer from ‘1 = very unfit’, to ‘5 = very fit’. A higher score represents a greater PPF. This question was used in previous studies [74].

Adolescents’ leisure time exercise was assessed using the Godin and Shepard Leisure-Time Exercise Questionnaire (LTEQ) [75]. The instrument measures individuals‘ leisure time exercise including light (e. g., yoga, bowling, easy walking), moderate (e.g., fast walking, easy bicycling, volleyball, easy swimming) and strenuous (e.g., running, football, vigorous long-distance bicycling) exercise at 15 min or more per session, over a typical week. Each frequency score is multiplied by a corresponding metabolic equivalent of task value, and the total leisure time activity score is calculated by this formula: light exercise is multiplied by 3, moderate exercise by 5, and strenuous exercise by 9, and the results were summarised. A higher score indicates higher PA in each of the three levels.

### 2.3. Statistical Analyses

According to the continually varying sample size approach to Monte Carlo power analysis, approximately 150 individuals are required to ensure a statistical power of at least 80% to detect the hypothesised indirect effect [76]. A power of 0.80 can be achieved for the simple model with one mediator with a sample size of 50–200.

The normality testing of study variables indicated that all study measures met normality assumptions except for the PA score. Next, study variables were compared across gender groups by independent samples t-test, and the effect sizes represented by Cohen’s d coefficients were calculated. Effect sizes above 0.2 were considered small, and those equal to or above 0.5 were considered moderate [77]. PA scores among boys and girls were compared using the Mann-Whitney U test. Correlations between study variables were tested by Pearson and Spearman (for the PA score) correlation coefficients, separately in boys and girls. Correlations between 0.1 and 0.3 were considered small, those above 0.3 and below 0.5 were considered moderate, and those equal to or above 0.5 were considered strong, with a significance level of <0.05 [77]. The internal consistency of the scales was tested by Cronbach’s α coefficients. A Cronbach’s α over 0.65 was considered adequate [78], and it should generally be noted that Cronbach’s α values are sensitive to the number of items included in the scale [79]. Preliminary statistical analysis was carried out using IBM SPSS Statistics v.28 (IBM Corp., Armonk, NY, USA).

The study variables were included in mediational structural equation models. The bootstrap approach was used to conduct mediation analyses with 5000 bootstrap samples drawn from the dataset to calculate indirect and direct effects, and bias corrected 95% CIs [80]. The 95% CIs for the coefficients calculated by bootstrapping methods were considered statistically significant if the confidence intervals did not include zero. Model fit was assessed using indices recommended by Hu and Bentler [81]: the normed model chi-square (χ2/df; values < 3.0 considered indicative of a good fit), the standardised root mean square residual (SRMR; values < 0.09 indicate a reasonable fit), the comparative fit index (CFI; values close to or >0.95 indicate an adequate fit), and the root mean square error of approximation (RMSEA) and its 90% CI (values close to 0.06 indicative of good fit and values up to 0.08 indicative of adequate fit). Mediation analysis was conducted using the Mplus v8.7 (Muthén & Muthén, Los Angeles, CA, USA).

## 3. Results

Table 1 presents the comparison of the study measures in gender groups. Boys demonstrated higher levels of PA, habitual exercise, motivation in PE, PPF, and body appreciation than girls. All the differences demonstrated small-to-medium effect sizes. There was no significant difference in perceived teacher support of autonomy during PE lessons across gender groups. Nevertheless, the correlations between all study measures (Table 2) were stronger in girls than boys, and demonstrated a positive direction and small-to-moderate effects.

A series of mediation models were run separately in boys and girls. In boys (the results are not presented in figures), PA habits mediated the association between motivation in PE and PPF, but the hypothesised model demonstrated a poor fit to data (CFI = 0.91, RMSEA = 0.12, SRMR = 0.05). In the moderated mediation model, PPF did not mediate the association between motivation in PE and body appreciation (*p* = 0.071), and the moderating effect of PA habits was not significant (*p* = 0.072).

In the girls’ mediation model (Figure 2), perceived teacher support of autonomy during PE lessons was positively associated with motivation in PE (β = 0.47, *p* < 0.001) and habitual exercise (β = 0.14, *p* = 0.013), while habitual exercise positively predicted PPF (β = 0.42, *p* < 0.001) in girls. Motivation in PE had positive effects on habitual exercise (β = 0.12, *p* = 0.036), PPF (β = 0.17, *p* < 0.001) and body appreciation (β = 0.28, *p* < 0.001). There was also a positive association between PPF and body appreciation in adolescent girls (β = 0.17, *p* = 0.001). The final girls’ model demonstrated a good fit to data: χ2 = 4.31, *p* = 0.230; df = 3; CFI = 0.995; SRMR = 0.021; RMSEA = 0.034 (90% CI = 0.01, 0.10).

Indirect (mediated) effects are presented in Table 3. Motivation in PE and PA habits mediated the associations between teacher support of autonomy during PE lessons, perceived physical fitness, and body appreciation. PPF was a mediator between PA habits and body appreciation.

A simplified moderated mediation model (triangle, Figure 3) in girls revealed positive associations between motivation in PE and PPF (β = 0.49, *p* = 0.001), between PPF and body appreciation (β = 0.37, *p* = 0.002), and between motivation in PE and body appreciation (β = 0.25, *p* < 0.001). Importantly there were two significant moderated effects in this model: PA habits moderated the associations between motivation in PE and PPF (β = −0.33, *p* = 0.027) and between PPF and body appreciation (β = −0.47, *p* = 0.042).

Importantly, in girls with the strongest habitual exercise (+1 SD), there were no effects from motivation in PE to PPF (B = 0.008, *p* = 0.103) or from PPF to body appreciation (B = 0.06, *p* = 0.504). On the contrary, adolescent girls with the lowest habitual exercise (−1 SD) had significant and stronger effects from motivation in PE to PPF (B = 0.024, *p* < 0.001) and from PPF to body appreciation (B = 0.27, *p* = 0.002) than girls with the mean habitual exercise (B = 0.016, *p* < 0.001 and B = 0.16, *p* = 0.015, accordingly). Figure 4 and Figure 5 present visualisations of PA habits moderating effects in the associations between motivation in PE and PPF and between PPF and body appreciation.

## 4. Discussion

We tested the SDT-based hypothetical model of teacher support of autonomy, student motivation for PE, and positive body image (operating as body appreciation). The present study makes a novel contribution to the literature, showing that teacher support of autonomy is associated with greater body appreciation in adolescent girls, and motivation for PE mediates this association. Specifically, the present study showed that students‘ PE classes in which autonomy is supported can provide an environment in which adolescents feel empowered and more positive about their physical fitness and body image. These findings agree with the results of a previous study emphasising the importance of associations between self-determined motivation, body satisfaction, and perception of sport as promoting the role of PE [12].

Previous studies showed that a more positive body image is associated with autonomous PA motivation [16,17], and the present study provides new knowledge about the associations between PE-related motivation and body image. It also tested positive body image as a consequence of motivation, as previously recommended by scholars [46]. The hypothetical model was only supported for girls; however, suggesting that the effect of PE motivation on positive body image might be stronger for adolescent girls compared to boys. In their study Kerner et al. [12] also found that associations between body satisfaction and self-determined motivation, and between PE motivation and the perceived sport-promoting role of PE, were stronger in adolescent girls than boys.

The PE environment might be stressful for students with body image concerns, or who are less athletically gifted, since students‘ bodies are exposed to social comparisons and public performance evaluation in PE [9]. Autonomy-supportive teachers use non-controlling language, provide a meaningful rationale, explore the life aspirations and values of students, share student perspectives, answer student questions, provide choices, show interest in students, and provide positive feedback [82]. Previous studies have showed that the behaviour of teachers in keeping a positive learning climate fulfils the BPNs of students (autonomy, competence and social relatedness), and might therefore help to resist body-related stress in PE and to keep a positive body image [44]. Participation in PE with an autonomy-supportive teaching climate might provide opportunities for students to experience their bodies performing successfully, and to feel self- and physical competence performing in front of others. Teacher support of autonomy and increasing autonomous motivation in PE might also help adolescent girls to perceive higher body acceptance from others, and to shift their attention from body appearance to the PA itself [83]. A longitudinal study showed that self-determined motivation and a low orientation around appearance are together responsible for less PA decline in adolescent students [49].

The results of this study confirm the main tenets of SDT, suggesting that teacher support of autonomy increases students‘ autonomous motivation for PE, and the latter is associated with various positive outcomes [44,84]. The present study adds important new knowledge that one of these outcomes might be a more positive body image. It seems that students, especially adolescent girls, who perceive higher autonomy support from their teachers, enjoy PE classes more and are less concerned about how their body is evaluated by others. Teacher support of autonomy is therefore important for the body image of adolescents, and teachers are recommended to use autonomy-supportive pedagogical strategies to promote positive body image in adolescents through PE. A previous systematic review concluded that interventions using cognitive dissonance, peer support, and psychoeducation provided evidence of improving body appreciation and body esteem in adolescent girls [85]. Our study showed that PE might be an important subject in which the positive body image of adolescent girls might be promoted. Since the study is cross-sectional, however, future studies with experimental designs should test our findings.

The results of the present study are also in line with TCM [62], suggesting that increased autonomous motivation in PE is associated with increased intentions and behaviour in associated contexts, such as leisure time PA [50,52,65,86]. The results of the present study showed that PE motivation is associated with elevated levels of PA habits. PA habits and PPF were mediators of PE motivation and positive body image in the present study. This finding might be explained by the hierarchical model EXSEM [8], suggesting that exercise promotes general self-esteem through elevated levels of self-efficacy, physical self-perception and increased self-worth. Previous studies showed that positive body image is associated with higher self-esteem in adolescent girls and boys [72], and that PPF is an important mediator in associations between exercise and body appreciation in adolescents [32]. Another possible explanation for these results might be based on the conception of body functionality [25]. Previous studies showed that sport-involved adolescents reported a more functional body image compared to adolescents not participating in sports, and a more positive attitude towards PE was associated with a more functional body image [87]. Body functionality is an important agent associated with positive body image [25]; however, we did not test functional body image, and we recommend including it in future studies.

For boys, teacher support of autonomy was associated with elevated body appreciation, however, autonomous motivation for PE was not. This might be explained by the fact that boys have greater body appreciation than girls [88], and their PA and PPF is significantly higher than that of girls [4,89]. Confronting the findings of previous research [90] in the present study, we found that boys reported higher autonomous motivation for PE than girls. We observed no significant differences in teacher support of autonomy between genders, but previous studies reported higher perceived autonomy support in boys [90]. The majority of studies use samples of both genders [44], and therefore, comparison of our results with other studies is limited.

The present study is cross-sectional, and we cannot answer the question—do adolescents with stronger PA habits and elevated levels of PPF have more positive experiences in PE, feel more supported by teachers and therefore experience more autonomous motivation for PE classes, or do adolescents have higher PPF and more positive body image as a result of positive educational climate and more intrinsic motivation in PE despite their PA levels? In trying to answer this question we tested the moderating role of PA habits in the associations between motivation for PE, PPF, and body appreciation. We found no moderating role of PA habits for adolescent boys. No moderating PA habits effect was observed for girls between PE motivation and body appreciation; however, PA habits moderated associations between motivation for PE and PPF in girls. Specifically, we found that associations between autonomous motivation for PA and PPF were highest in the group with the weakest PA habits. In other words, the effect of autonomous motivation for PE on PPF is strongest in girls with weaker PA habits, suggesting that higher positive body image perception might be a result of autonomous motivation for PE.

Finally, our study showed that adolescent girls with the lowest PA habits reported the highest associations between PPF and body appreciation. Specifically, our results show that the girls who have the strongest PA habits possibly have greater confidence in their physical abilities, and therefore increased PPF has no effect on their positive body image. Conversely, an increase in PPF in girls with the weakest PA habits has the strongest positive body image-promoting effect. PA interventions in the physical self-concept of adolescents showed that greater effects were achieved for those with lower initial levels of PA and physical self-perceptions, since they had wider margins for improvement [31]. These results suggest that increasing self-confidence through positive PPF might be an effective strategy, especially for girls with the lowest PA.

### 4.1. Strengths and Limitations

The present study has important limitations. First, we did not assess BPNs, which are important mediators between teacher support of autonomy and student motivation for PE [44]. In the present study, we tested only teachers‘ autonomy support, however, support for competence and relatedness might also be important for the development of self-determined motivation for PE and positive body image. Secondly, we did not assess functional body image, which might help to more deeply explain possible mechanisms in the associations between PE motivation, PPF, and body appreciation [25]. Future studies might also benefit from testing the possible role of self-objectification in the associations between PE motivation and body appreciation [83]. The generalization of the results might be limited since the present study was implemented in a relatively small and non-probability sample of Lithuanian adolescents. Future studies in larger samples and different cultures should test our findings. In future studies, we also recommend objectively measuring PA since self-reported leisure-time PA might be overestimated by adolescents. Finally, the present study is cross-sectional, and does not enable us to understand the directions of the tested associations. Future studies with other than cross-sectional designs should test our findings. Nevertheless, the present study has important strengths that are worth mentioning. First, it is one of the first studies testing associations between motivational processes in PE and positive body image as an outcome of motivation. We tested PE motivation-related processes in different genders, which has been previously recommended by scholars [44].

### 4.2. Practical Implications

Several practical recommendations might be drawn from the results of the present study. Our study showed that PE might be an important subject in which the positive body image of adolescent girls might be promoted. PE teachers are strongly recommended to use autonomy-supportive pedagogical methods, and to promote the autonomous motivation of students, and especially adolescent girls with weaker PA habits. When students experience autonomous motivation, they exercise more outside school (boys), feel more confident in their physical fitness, and express a more positive body image (girls). Teacher support of autonomy and autonomous motivation for PE are important for all students, but the highest effect of motivation for PE on PPF might be achieved in adolescent girls with the weakest PA habits.

## 5. Conclusions

The present study extends the previous literature based on SDT and explores the associations between PE motivation and body image in adolescents. The results of the study showed that autonomous motivation in PE is an important factor related to more positive body image in adolescents. In adolescent girls, autonomous motivation for PE was the mediator between teacher support of autonomy and positive body image. The associations between intrinsic motivation for PE and positive body image were mediated by PPF on one hand, and through PA habits and PPF on the other hand of the structural equation model (in girls). PA habits moderated associations between PE motivation and PPF. Associations between PE motivation and PPF were stronger in girls with the weakest PA habits, compared to girls with mean PA habits. These results suggest that PE subject is important for promoting positive body images among students. PE teachers are recommended to use pedagogical strategies that promote students‘ self-determined motivation for PE. This might be an effective strategy for promoting physical self-perception and positive body image in adolescent girls, especially those with weak PA habits.

## Figures and Tables

**Figure 1 ijerph-20-00464-f001:**
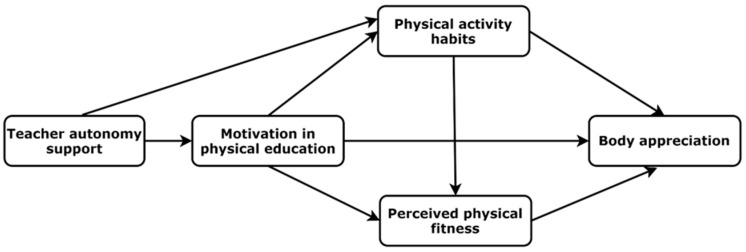
Hypothetical model of the associations between study measures.

**Figure 2 ijerph-20-00464-f002:**
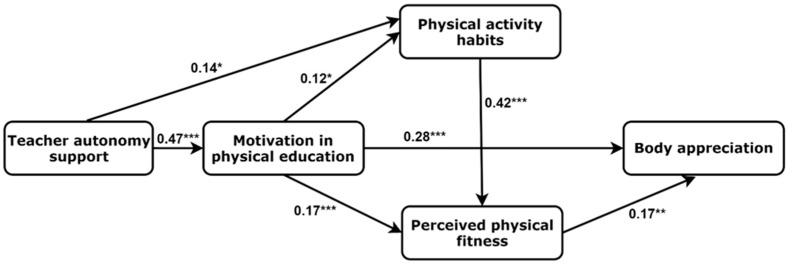
Hypothetical model of the associations between study measures. * *p* < 0.05; ** *p* < 0.01; *** *p* < 0.001.

**Figure 3 ijerph-20-00464-f003:**
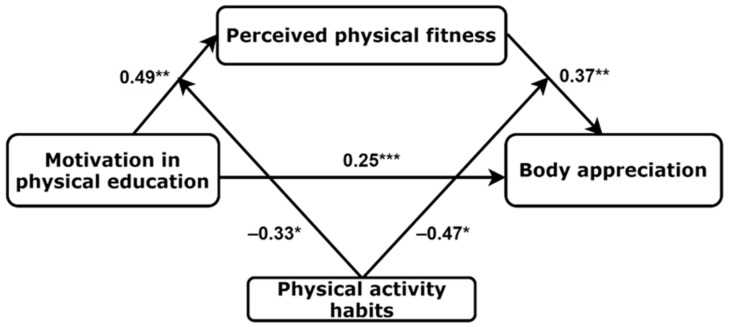
Physical activity habits moderate the associations between motivation in physical education, perceived physical fitness, and body appreciation in adolescent girls (n = 371). * *p* < 0.05; ** *p* < 0.01; *** *p* < 0.001.

**Figure 4 ijerph-20-00464-f004:**
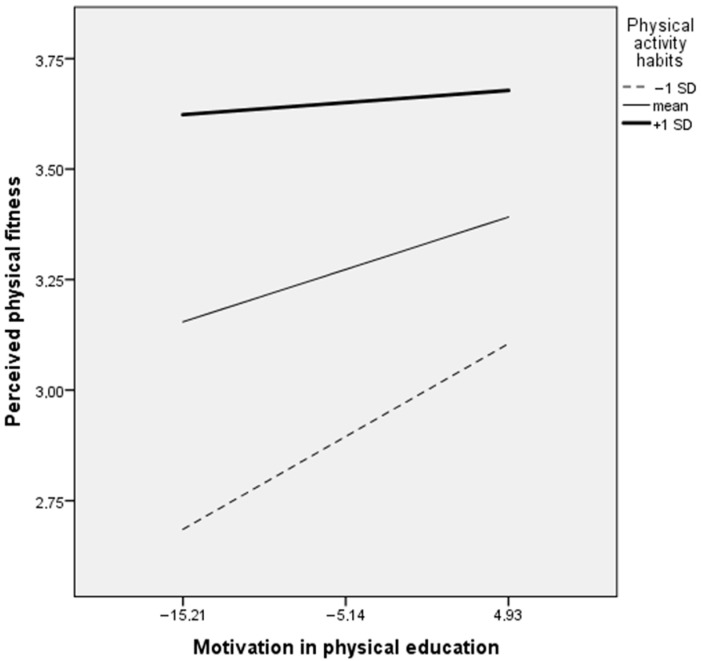
Visualisation of physical activity habits moderating effect in the association between motivation in physical education and perceived physical fitness (n = 371).

**Figure 5 ijerph-20-00464-f005:**
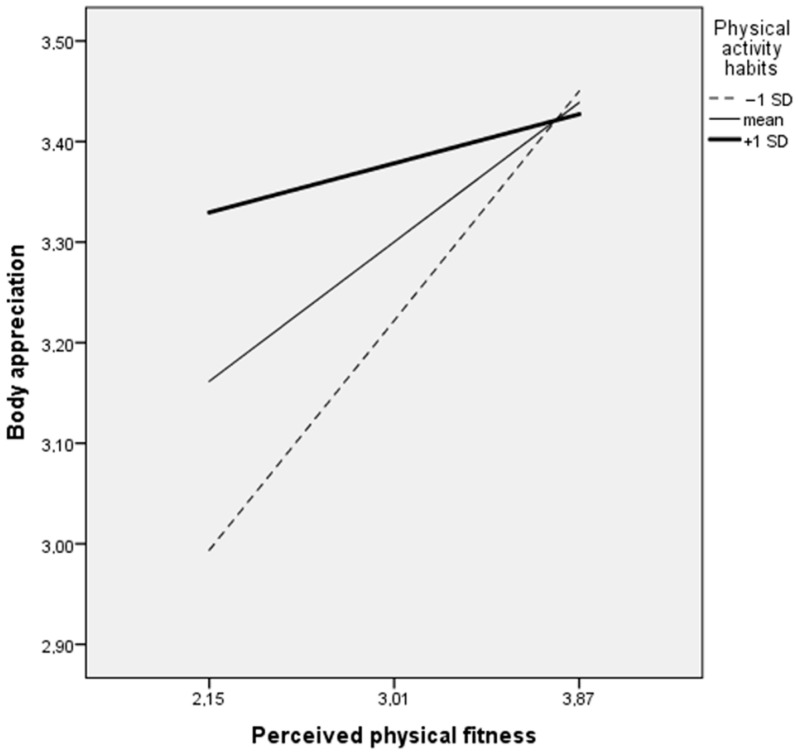
Visualisation of physical activity habits moderating effect in the association between perceived physical fitness and body appreciation (n = 371).

**Table 1 ijerph-20-00464-t001:** The comparison of the study measures between boys and girls, m (SD), n = 715.

Study Measures	Range	Boys, n = 344	Girls, n = 371	Cohen‘ *d*	*p*
Leisure-time exercise (LTEQ)	0–395	83.3 (54.7)	58.4 (41.5)	0.51	<0.001
Physical activity habits	1–7	4.3 (1.3)	3.7 (1.2)	0.42	<0.001
Perceived teacher support of autonomy (LCQ)	1–7	4.8 (1.3)	4.8 (1.3)	-	0.872
Motivation for PE (PLOC-R)	−32.6–13.7	−3.6 (9.5)	−6.6 (10.3)	0.31	<0.001
Perceived physical fitness	1–5	3.5 (0.9)	3.0 (0.9)	0.56	<0.001
Positive body image (BAS-2)	1–5	3.8 (0.9)	3.3 (1.0)	0.53	<0.001

m = mean, SD = standard deviation, LTEQ = Leisure-Time Exercise Questionnaire, LCQ = Learning Climate Questionnaire, PE = physical education, PLOC-R = Perceived Locus of Causality in PE Scale, revised version, BAS-2 = Body Appreciation Scale 2.

**Table 2 ijerph-20-00464-t002:** Correlations between study measures in boys and girls (n = 715).

Study Measures	LTEQ	PAH	LCQ	PLOQ-R	PPF	BAS-2
Leisure-time exercise (LTEQ)	1.00	0.32 **	0.05	0.10	0.19 **	0.06
Physical activity habits (PAH)	0.25 **	1.00	0.20 **	0.19 **	0.45 **	0.18 **
Perceived teacher support of autonomy (LCQ)	0.003	0.20 **	1.00	0.47 **	0.12 **	0.21 **
Motivation for PE (PLOC-R)	−0.03	0.17 **	0.33 **	1.00	0.25 **	0.32 **
Perceived physical fitness (PPF)	0.15 **	0.36 **	−0.004	0.08	1.00	0.24 **
Positive body image (BAS-2)	0.13 *	0.22 **	0.19 **	0.10	0.33 **	1.00

Correlations for boys are presented in the lower diagonal, for girls—in the upper; * *p* < 0.05; ** *p* < 0.01; LTEQ = Leisure-Time Exercise Questionnaire, LCQ = Learning Climate Questionnaire, PE = physical education, PLOC-R = Perceived Locus of Causality in PE Scale, revised version, BAS-2 = Body Appreciation Scale 2.

**Table 3 ijerph-20-00464-t003:** Standardised indirect effects in girls’ mediation model, n = 371.

Paths	β	95% CI LB	95% CI LB	*p*
TAS → MPE → PAH	0.06	−0.004	0.12	0.075
TAS → MPE → PAH → PPF	0.16	0.10	0.23	<0.001
TAS → MPE → PAH → PPF → BA	0.16	0.11	0.21	<0.001
MPE → PAH → PPF	0.05	−0.003	0.11	0.070
MPE → PAH → PPF → BA	0.04	0.01	0.07	0.008
PAH → PPF → BA	0.07	0.02	0.12	0.003

CI = confidence interval, LB = lower bound, UB = upper bound; TAS = teacher support of autonomy, MPE = motivation in PE, PAH = physical activity habits, PPF = perceived physical fitness, BA = body appreciation.

## Data Availability

The dataset generated and analysed during the current study is not publicly available, but is available from the corresponding author upon reasonable request.

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
