# Peer review of "Association between Motivation in Physical Education and Positive Body Image: Mediating and Moderating Effects of Physical Activity Habits"

_ijerph, 2022, doi:10.3390/ijerph20010464_

Round 1
Reviewer 1 Report
The review comments are listed as follows.
1. Although Figure1 shows the hypothetical model in this study; however, the authors should clearly describe the hypothesis of this study in the manuscript.
2. Some abbreviations appear in the manuscript should have their full text.
3. It is better to use the same terms in Table 1 and the paragraph that described the study results related to Table1.
4. The authors use Figure 2 to show the girls’ mediation model; why do not the authors use another figure to show the boy’s mediation model?
5. The authors should explain the reasons why they only focus on girl’s assessment results in the latter part of Section 3.
6. The authors also should use the adolescent boys’ data to have more explorations in Section 3.
7. Line 414 mentions “Figures 4 and 6 present”. However, there is no Figure 6 in the manuscript.
8. The authors should have more explanations to explore the implications of Figure 4 and Figure 5.
9. The citation of literature No. 80 cannot be found in the manuscript.
Author Response
Thank you for your time reviewing our paper and for your valuable comments. All changes made in the text are highlighted in a blue font.
- Although Figure 1 shows the hypothetical model in this study; however, the authors should clearly describe the hypothesis of this study in the manuscript.
Hypotheses were developed (p. 5-6, lines 247-253).
- Some abbreviations appear in the manuscript should have their full text.
The text was double-checked for abbreviations.
- It is better to use the same terms in Table 1 and the paragraph that described the study results related to Table 1.
Terms were aligned in Table 1 and the Instruments section.
- The authors use Figure 2 to show the girls’ mediation model; why do not the authors use another figure to show the boy’s mediation model?
In boys, the hypothesized model showed a poor fit to data, however, information regarding boys is presented in lines 393-397, p. 9.
- The authors should explain the reasons why they only focus on girl’s assessment results in the latter part of Section 3. And 6. The authors also should use the adolescent boys’ data to have more explorations in Section 3.
Information regarding boys is presented in lines 393-397, p. 9. Other analysis of boys‘ results has no meaning in the present sample.
- 7. Line 414 mentions “Figures 4 and 6 present”. However, there is no Figure 6 in the manuscript.
Corrected, thank you.
- The authors should have more explanations to explore the implications of Figure 4 and Figure 5.
Practical implications for Fig. 4 and 5 are included in the section Practical Implications where we state that support for autonomy and internal motivation might have a higher effect on physical fitness perception in girls with low physical habits compared to those with high habits and teachers are recommended to show more support of autonomy for the girls with low physical activity habits.
- The citation of literature No. 80 cannot be found in the manuscript.
This reference is in Statistical analysis section, line 364, p. 8.
Reviewer 2 Report
This paper is sound, concise and well written. The results are relevant and the conclusions of the study are robust. I have very minor comments:
1) Please note that the scientific way to express the 16.00 (SD = 0.79) for girls 15
and 15.99 (SD = 0.75) shoul dbe 16.0 (SD=0.8) and 16.0 (SD=0.8).
2) In the methodology add something like:
As the original version of the questionnaires was in English, they were translated into Lithuanian. A speaker whose first language (L1) is English translated the questionnaires back into English after they had first been translated into Lithuanian by a speaker whose first language (L1) is Lithuanian. The translation procedure turned up no significant problems.
3) In Figure 4, please do not use commas. I guess the value -15,21 is wrong. Also in Figure 5.
Author Response
This paper is sound, concise and well written. The results are relevant and the conclusions of the study are robust. I have very minor comments:
Thank you for your time reviewing our paper and for your valuable comments. Also, thank you for the positive feedback.
All changes made in the text are highlighted in a blue font.
- Please note that the scientific way to express the 16.00 (SD = 0.79) for girls 15
and 15.99 (SD = 0.75) should be 16.0 (SD=0.8) and 16.0 (SD=0.8).
Thank you, corrected.
- In the methodology add something like:
As the original version of the questionnaires was in English, they were translated into Lithuanian. A speaker whose first language (L1) is English translated the questionnaires back into English after they had first been translated into Lithuanian by a speaker whose first language (L1) is Lithuanian. The translation procedure turned up no significant problems.
All Lithuanian versions of instruments were validated in previous studies, and a detailed description of translation procedures was presented there. Therefore, we did not present this information in the present study.
- In Figure 4, please do not use commas. I guess the value -15,21 is wrong. Also in Figure 5.
All the values are calculated and provided automatically by the SPSS Syntax file. They are correct. Motivation in PE includes negative values as well. The description of the scoring of the motivation in PE is provided in the Methods section, lines 295-311.
Unfortunately, for the Figures 4 and 5 there is an option only to make changes in the X axis, but not in the Y. To keep the same formatting in the Figures, we left comas.
Reviewer 3 Report
First of all, I would like to thank you for sending me this article for review and congratulate the authors for their initiative in this highly relevant research in the field of health of the adolescents.
The article presents the objective of we assessed the relationships between teacher support of autonomy, student motivations for PE, and positive body image, in a sample of Lithuanian adolescents.
The article shows a cross-sectional design and shows mediating and moderating effect analysis which is considered very appropriate for this type of article.
It is recommended to use keywords that were not included in the title of the article.
The introduction is wide, deep and adequate, providing a large number of appropriate and updated references.
The section on study participants and procedures includes a lot of relevant and necessary information. However, it is not stated that the Strobe Statement for cross-sectional studies is followed. It is recommended that this statement be included. In addition, it is recommended that this guide be consulted to include those aspects that have not been included.
The study design is indicated in the abstract, but is not indicated in the study. The design should be indicated in the study participant and procedure agreement.
Place, dates, recruitment and evaluation periods should be indicated.
The inclusion and exclusion criteria of the participants are not indicated. They should be indicated in detail.
Sample and power size should be included. The sample is not too large but this value is important.
The conditions under which the subjects were measured should be described in more detail.
Indicate who performed the measurements and the number of investigators, information given to participants, etc. Indicate whether the investigators who administered the questionnaires were always the same and whether they were instructed in their explanation.
Potential sources of bias should be included.
Author Response
Thank you for your time reviewing our paper and for your valuable comments. All changes made in the text are highlighted in a blue font.
First of all, I would like to thank you for sending me this article for review and congratulate the authors for their initiative in this highly relevant research in the field of health of the adolescents.
The article presents the objective of we assessed the relationships between teacher support of autonomy, student motivations for PE, and positive body image, in a sample of Lithuanian adolescents.
The article shows a cross-sectional design and shows mediating and moderating effect analysis which is considered very appropriate for this type of article.
Thank you for your positive feedback.
It is recommended to use keywords that were not included in the title of the article.
Keywords were revised, thank you.
The introduction is wide, deep and adequate, providing a large number of appropriate and updated references.
Thank you.
The section on study participants and procedures includes a lot of relevant and necessary information. However, it is not stated that the Strobe Statement for cross-sectional studies is followed. It is recommended that this statement be included. In addition, it is recommended that this guide be consulted to include those aspects that have not been included.
Thank, you, the sentence of use Strobe statement was included and the manuscript was double-checked.
The study design is indicated in the abstract, but is not indicated in the study. The design should be indicated in the study participant and procedure agreement.
The information is included in the Methods section as well.
Place, dates, recruitment and evaluation periods should be indicated. The inclusion and exclusion criteria of the participants are not indicated. They should be indicated in detail.
Thank you for this important comment. We have included this information to the section 2.1.
Sample and power size should be included. The sample is not too large but this value is important.
This information was added to the section 2.3, Statistical analysis:
According to the continually varying sample size approach to Monte Carlo power analysis, approximately 150 individuals are required to ensure a statistical power of at least 80% to detect the hypothesized indirect effect [76]. A power of 0.80 can be achieved for the simple model with one mediator with a sample size of 50–200.
The conditions under which the subjects were measured should be described in more detail.
We have included this information. Study participants filled-in the survey during the theoretical PE lessons.
Indicate who performed the measurements and the number of investigators, information given to participants, etc. Indicate whether the investigators who administered the questionnaires were always the same and whether they were instructed in their explanation.
Thank you, this information was included, and PE teachers were instructed prior to the survey. Seven PE teachers participated in the research.
Potential sources of bias should be included.
Thank you for this important comment. We added this information to the study limitations:
The generalization of the results might be limited since the present study was implemented in a relatively small and non-probability sample of Lithuanian adolescents. Future studies in larger samples and different cultures should test our findings. In future studies, we also recommend objectively measuring PA since self-reported leisure-time PA might be overestimated by adolescents.
Round 2
Reviewer 1 Report
All comments are revised. However, the citation symbol in line 367 is incorrect, please fix it.
Author Response
Dear Reviewer,
The citation symbol has been fixed, thank you.
Reviewer 3 Report
Thank your for the review. Congratulations.
Author Response
Thank you!